# The transcription factor Pou3f1 promotes neural fate commitment via activation of neural lineage genes and inhibition of external signaling pathways

Qingqing Zhu[1,2†], Lu Song[1†], Guangdun Peng[1], Na Sun[3], Jun Chen[1], Ting Zhang[1], Nengyin Sheng[1], Wei Tang[1], Cheng Qian[1], Yunbo Qiao[1], Ke Tang[4], Jing-Dong Jackie Han[3], Jinsong Li[1], Naihe Jing[1]*

[1]State Key Laboratory of Cell Biology, Institute of Biochemistry and Cell Biology, Shanghai Institutes for Biological Sciences, Chinese Academy of Sciences, Shanghai, China; [2]Department of Neurosurgery, West China Hospital, Sichuan University, Sichuan, China; [3]Key Laboratory of Computational Biology, CAS-MPG Partner Institute for Computational Biology, Shanghai Institutes for Biological Sciences, Chinese Academy of Sciences, Shanghai, China; [4]Institute of Life Science, Nanchang University, Nanchang, Jiangxi, China

**Abstract** The neural fate commitment of pluripotent stem cells requires the repression of extrinsic inhibitory signals and the activation of intrinsic positive transcription factors. However, how these two events are integrated to ensure appropriate neural conversion remains unclear. In this study, we showed that Pou3f1 is essential for the neural differentiation of mouse embryonic stem cells (ESCs), specifically during the transition from epiblast stem cells (EpiSCs) to neural progenitor cells (NPCs). Chimeric analysis showed that Pou3f1 knockdown leads to a markedly decreased incorporation of ESCs in the neuroectoderm. By contrast, Pou3f1-overexpressing ESC derivatives preferentially contribute to the neuroectoderm. Genome-wide ChIP-seq and RNA-seq analyses indicated that Pou3f1 is an upstream activator of neural lineage genes, and also is a repressor of BMP and Wnt signaling. Our results established that Pou3f1 promotes the neural fate commitment of pluripotent stem cells through a dual role, activating internal neural induction programs and antagonizing extrinsic neural inhibitory signals.

*For correspondence: njing@ sibcb.ac.cn

†These authors contributed equally to this work

## Introduction

Early vertebrate development is the process by which unrestricted pluripotent stem cells progressively make lineage fate choices. Central to cell allocation is gastrulation, during which the epiblast responds to secreted signals and generates three primary germ layers (*Lu et al., 2001*). In early mouse embryos, gastrulation initiates at embryonic day (E) 6.5. Posterior epiblast cells ingress through the primitive streak to form the mesoderm and endoderm, whereas the cells that remain in the anterior part of the epiblast form the ectoderm (*Tam and Loebel, 2007*). Then, a portion of the anterior ectoderm is specified to adopt the neural fate and subsequently, develops into the neuroectoderm, forming a plate-shaped structure called the neural plate at approximately E7.5 (*Tam and Zhou, 1996*).

Previous studies have indicated that neural fate specification from embryonic ectoderm occurs autonomously in the absence of inhibitory signals such as bone morphogenetic proteins (BMPs) and Wnts (*Munoz-Sanjuan and Brivanlou, 2002*; *Stern, 2005b*). In early *Xenopus*, chick, and mouse embryos, BMP and Wnt signals prevent neural conversion and contribute to non-neural fates such as

**eLife digest** After an egg has been fertilized, it undergoes a series of divisions to produce a ball of cells known as a blastocyst. The cells within the blastocyst are pluripotent stem cells, which have the potential to become many different types of cell. After a few days, the stem cells organize into three layers—an innermost layer called the endoderm, a middle layer of mesoderm, and an outer layer of ectoderm—that ultimately give rise to different types of tissues.

The brain and nervous system are formed from cells in the neuroectoderm, which is part of the ectoderm. Now, Zhu et al. have shown that a transcription factor called Pou3f1 triggers stem cells within a region of the ectoderm to turn into neural progenitor cells, thereby generating the neuroectoderm. These neural progenitor cells then go on to become neurons and glial cells that make up the brain and nervous system.

Using a virus to reduce levels of Pou3f1 in embryonic stem cells grown in a dish led to a drop in the number of stem cells that committed to neural progenitor cells. Overexpressing Pou3f1 in the stem cells restored the number of neural progenitor cells. Together these results showed that Pou3f1 is both necessary and sufficient for the conversion of embryonic stem cells into future neurons and glia.

The same result was seen when embryonic stem cells containing either reduced or elevated levels of Pou3f1 were injected into 2.5-day-old mouse blastocysts, which were then implanted into surrogate females. The resulting embryos comprised some cells with normal levels of Pou3f1, and others with either too little or too much. Cells with elevated Pou3f1 mostly became neural progenitors, whereas those with reduced levels rarely did so. Gene expression studies revealed that Pou3f1 promoted the formation of neural progenitor cells by activating the expression of pro-neuronal genes inside the stem cells, and by blocking anti-neuronal pathways called Wnt/BMP signaling cascades initiated outside the cells.

By revealing the two roles of Pou3f1, Zhu et al. have increased our understanding of one of the earliest stages of nervous system development. Further work is required to determine exactly how Pou3f1 exerts its effects and, in particular, whether it performs its two roles simultaneously or in sequence.

epidermal differentiation and primitive streak formation (*Winnier et al., 1995*; *Hemmati-Brivanlou and Melton, 1997*; *Liu et al., 1999*; *Wilson et al., 2001*). BMP and Wnt inhibition in the prospective neural ectoderm is essential for proper neural development (*Wilson and Edlund, 2001*). In mouse embryonic stem cells (ESCs), BMP and Wnt signals are required for self-renewal and readily repress neural differentiation partially through their targets, such as *Id1*, *Id2*, and *Myc* (*ten Berge et al., 2011*; *Varlakhanova et al., 2010*; *Ying et al., 2003*; *Zhang et al., 2010a*). BMP and Wnt antagonists have been utilized to generate neural lineage cells in mouse or human ESCs (*Blauwkamp et al., 2012*; *Chambers et al., 2009*; *Gratsch and O'Shea, 2002*; *Watanabe et al., 2005*).

In addition to extrinsic signaling pathways, neuroectoderm specification is also controlled by the sequential activation of intrinsic neural fate-promoting factors. Sox2, which is an ESC pluripotency-maintenance factor, plays an important role in ESC neural differentiation, indicating that Sox2 is a neural lineage-poised factor (*Thomson et al., 2011*). Zic2 and Otx2 are also involved in epiblast stem cell (EpiSC) neural conversion (*Iwafuchi-Doi et al., 2012*). Recently, Zfp521 was identified as an intrinsic factor that promotes the progression of early neural development (*Kamiya et al., 2011*). Studies concerning these neural fate-promoting factors have partially revealed the internal mechanism of early neural development. However, how these neural factors are activated during neural fate commitment remains unclear. Moreover, considering the importance of the effect of extrinsic signals on the neural fate decision, it remains unclear whether the inhibition of extrinsic signals and activation of internal factors are regulated separately or are integrated by a single determinant.

POU family transcription factors play important roles in the development of the nervous system (*Veenstra et al., 1997*). Pou3f1 (also known as Oct6, Tst1, or as SCIP) has been reported as the earliest expressed POU III family member in mouse embryo development (*He et al., 1989*; *Monuki et al., 1989*; *Meijer et al., 1990*; *Suzuki et al., 1990*). During gastrulation, *Pou3f1* expression is observed in the chorion and in the anterior epiblast (*Zwart et al., 1996*). As embryonic development proceeds,

*Pou3f1* expression becomes restricted to central nervous tissues and is detectable in the midbrain and in the forebrain (*He et al., 1989*; *Zwart et al., 1996*). Pou3f1 has also been documented as a crucial regulator of the myelination of Schwann cells in the peripheral nervous system (*Bermingham et al., 1996*; *Jaegle et al., 1996*). In vitro, the rapid increase of *Pou3f1* mRNA in retinoic acid-induced neural differentiation of P19 cells suggests that Pou3f1 may be functionally associated with neural fate commitment (*Meijer et al., 1990*). Recent reports have proposed that Pou3f1 might be a potential regulator associated with early neural development (*Kamiya et al., 2011*; *Iwafuchi-Doi et al., 2012*; *Yasuhara et al., 2013*). However, whether Pou3f1 is involved in the neural initiation of pluripotent stem cells remains elusive, and the underlying mechanism requires further investigation.

In this study, we show that Pou3f1 is necessary and sufficient for the neural fate commitment of ESCs and of EpiSCs. In chimeric mice, Pou3f1-knockdown cells display suppressed neuroectoderm distribution. Conversely, ESCs with Pou3f1 overexpression preferentially contribute to the neuroectoderm but not to other lineages. We further demonstrate that Pou3f1 promotes the neural fate commitment of pluripotent stem cells through the activation of intrinsic neural lineage genes and through the inhibition of extrinsic BMP and Wnt signals.

## Results

### Pou3f1 is essential for ESC neural differentiation

We previously established an efficient system to induce ESC neural differentiation in serum-free medium (*Zhang et al., 2010a*). To investigate neural conversion mechanisms, we performed a microarray-based screening and identified *Pou3f1* as one of the genes significantly up-regulated during pluripotent stem cell neural differentiation. Pou3f1 was moderately expressed in ESCs. The highest levels were observed from days 2–4 upon neural differentiation, and then the expression of Pou3f1 declined (*Figure 1A*, *Figure 1—figure supplement 1A*). Gene expression profiling indicated that the *Pou3f1* expression peak occurred between the epiblast marker *Fgf5* and the neural stem cell marker *Sox1* (*Figure 1A*, *Figure 1—figure supplement 1A*). This result suggests that Pou3f1 might play a role in ESC neural differentiation.

To test this hypothesis, a lentivirus-mediated knockdown strategy was utilized to diminish Pou3f1 expression. Two shRNAs (KD1 and KD3) targeting the *Pou3f1* 3' UTR region efficiently decreased *Pou3f1* expression in ESCs to approximately 50% and 30%, respectively (*Figure 1—figure supplement 1B*). The control (Ctrl) and Pou3f1-KD1/3 ESCs exhibited comparable expression levels of the pluripotency markers (*Figure 1—figure supplement 1C*). After differentiating these ESC lines in serum-free medium, the transcripts of the neural markers *Sox1*, *Pax6*, and *Tuj1* were reduced in Pou3f1-KD1/3 cells (*Figure 1B*). Immunocytochemical assays confirmed the reduced percentage of Sox$^+$/Oct4$^-$, Pax6$^+$ NPCs, and Tuj1$^+$ neurons from Pou3f1-KD1/3 ESCs (*Figure 1C,D*). Moreover, unbiased ESC differentiation in serum-containing medium revealed that the expression of mesoderm (*T* and *Flk1*), endoderm (*Gata4* and *Gata6*), and epidermal (*Ck18* and *Ck19*) markers was unaltered after *Pou3f1* knockdown (*Figure 1—figure supplement 1D*). These results suggest that Pou3f1 is selectively required for the neural differentiation of ESCs.

Because most POU III proteins exhibit extensive functional equivalence (*Andersen and Rosenfeld, 2001*; *Jaegle et al., 2003*; *Friedrich et al., 2005*), we wanted to determine whether other POU III proteins, such as Brn1 and Brn2, are similarly involved in ESC neural fate commitment. We examined the *Brn1* and *Brn2* expression profiles, and determined that these proteins are up-regulated in ESC serum-free neural differentiation after day 5, following the peak of *Pou3f1* expression (*Figure 1—figure supplement 2A*). Interestingly, compared with the control, *Brn2*, but not *Brn1*, expression was enhanced in Pou3f1-KD1/3 cells (*Figure 1—figure supplement 2B*). When the expression of *Pou3f1* and *Brn2* was simultaneously reduced by lentivirus-mediated shRNAs, the expression of the neural marker genes *Sox1*, *Pax6*, and *Nestin* decreased more dramatically (*Figure 1—figure supplement 2D*), although *Brn1* expression was not affected (*Figure 1—figure supplement 2C*). Together, these results suggest that *Brn2*, which is a POU III family member, compensates for Pou3f1 depletion.

To determine whether Pou3f1 is sufficient to promote the neural differentiation of ESCs, stable Pou3f1-overexpressing ESCs were generated. Compared with the control, the constitutive expression of Pou3f1 notably enhanced the expression of NPC and the neuron markers during serum-free differentiation, particularly at day 4 (*Figure 1—figure supplement 3*). Single cell suspensions from EBs at various days were replated in N2 medium for neuronal differentiation. Many Tuj1$^+$ neurons emerged

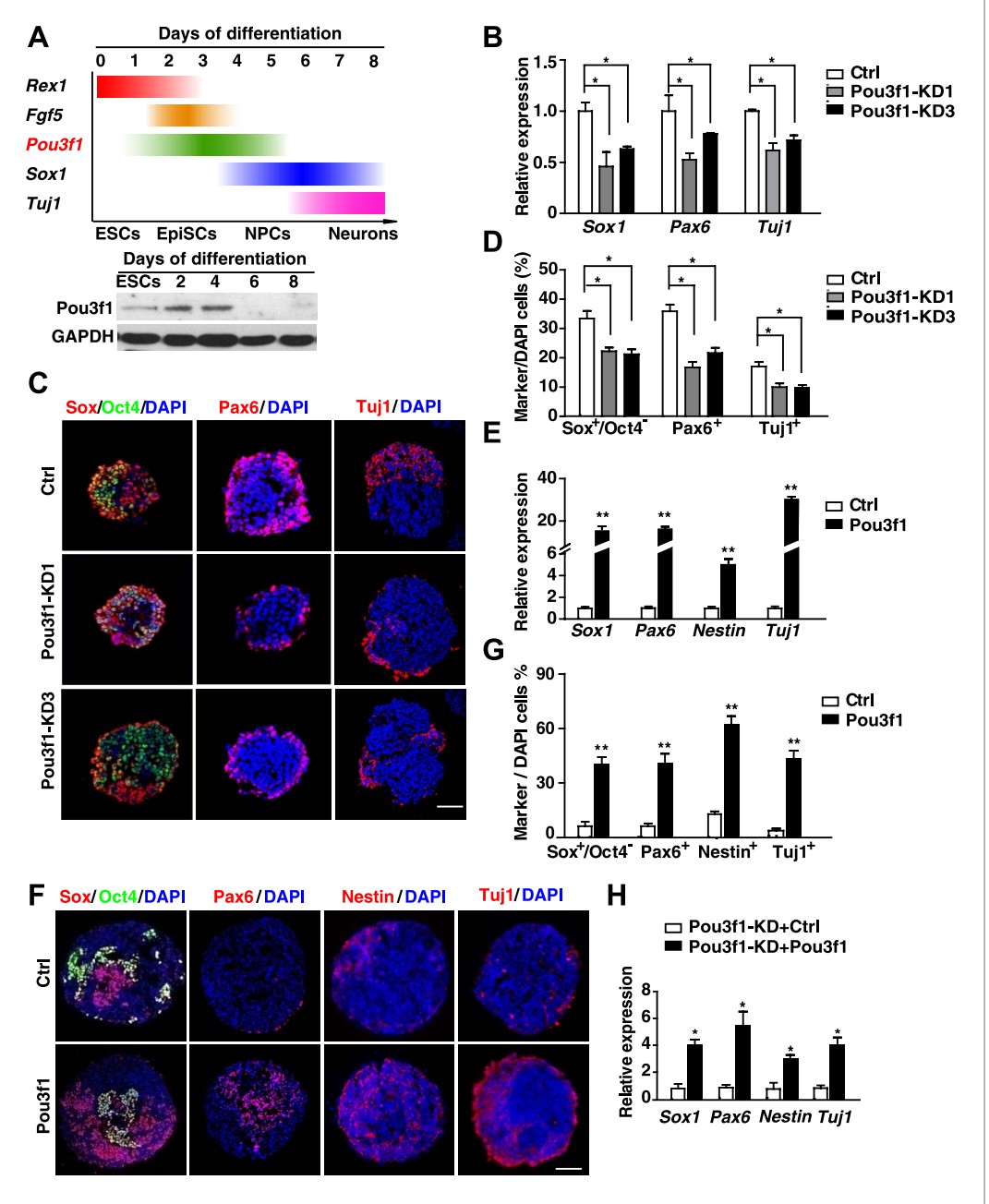

**Figure 1**. Pou3f1 is essential for ESC neural differentiation. (**A**) Schematic expression profiles of Pou3f1 and of several key marker genes during ESC neural differentiation in serum-free medium. *Rex1*, ESC marker; *Fgf5*, EpiSC marker; *Sox1*, NPC marker; *Tuj1*, neuron marker. Detection of Pou3f1 protein expression during ESC neural differentiation by Western blotting. (**B**) Gene expression levels in control-ESCs (Ctrl) and in Pou3f1-knockdown ESCs (Pou3f1-KD1, Pou3f1-KD3) at neural differentiation day 4 were determined by Q-PCR. Three independent experiments were performed. (**C**) Immunocytochemical assays of Sox/Oct4, Pax6, and Tuj1 in day 4 EBs described in **B**. DNA is stained with DAPI. Scale bars: 50 µm. (**D**) Statistical analysis of Sox$^+$/Oct4$^-$, Pax6$^+$, and Tuj1$^+$ cells in **C**. (**E**) Gene expression levels in control-ESCs and inducible Pou3f1-overexpressing (Pou3f1-OE) ESCs at unbiased differentiation (10%FBS) day 8 were determined by Q-PCR. Dox (2 µg/ml) was added for 8 days. (**F**) Immunocytochemical assays of Sox/Oct4, Pax6, Nestin, and of Tuj1 in day 8 EBs described in **E**. Scale bars, 50 µm. (**G**) Statistical analysis of Sox$^+$/Oct4$^-$, Pax6$^+$, and Tuj1$^+$ cells in **F**. (**H**) Pou3f1-knockdown ESCs were transfected with control or with Pou3f1-overexpressing lentiviruses. Gene expression levels at neural differentiation day 4 were determined by Q-PCR. The values represent the mean ± SD for **B**, **D**, **E**, **G**, and for **H**. (*p<0.05; **p<0.01).

*Figure 1. Continued on next page*

*Figure 1. Continued*

The following figure supplements are available for figure 1:

**Figure supplement 1**. *Pou3f1*-knockdown ESCs could differentiate into non-neural cell lineages.

**Figure supplement 2**. *Brn2* could compensate for the *Pou3f1* depletion during ESC neural fate commitment.

**Figure supplement 3**. Overexpression of *Pou3f1* accelerates ESC neural differentiation in serum-free condition.

**Figure supplement 4**. *Pou3f1* promotes neural differentiation in a cell-autonomous manner.

from stable Pou3f1-overexpression ESCs at day 4, 2 days earlier than the control ESCs (*Figure 1—figure supplement 3C*, d). These results demonstrate that neural differentiation was accelerated by Pou3f1 overexpression under serum-free conditions. To exclude the influences of Pou3f1 overexpression on the ESC state, doxycycline (Dox)-inducible Pou3f1-overexpressing ESCs were generated (*Figure 1—figure supplement 4A*). As expected, the Dox-induced overexpression of *Pou3f1* strongly enhanced ESC neural differentiation in serum-containing medium, which was accompanied by the increased expression of the neural markers Sox1, Pax6, Nestin, and Tuj1 in both quantitative polymerase chain reaction (Q-PCR) and immunostaining assays (*Figure 1E–G*). Moreover, the decreased neural marker expression in Pou3f1-depleted ESCs was restored by the overexpression of a *Pou3f1* coding sequence (CDS) lacking the 3′ UTR (*Figure 1H*). Cell aggregation assays were performed by co-culturing wild-type ESCs with either GFP-labeled control or Pou3f1-overexpressing ESCs in serum-free medium. The neural differentiation of wild-type ESCs was not affected by Pou3f1-overexpressing ESCs in the culture system (*Figure 1—figure supplement 4B*), indicating that Pou3f1 promoted neural differentiation cell-autonomously. Taken together, these results suggest that Pou3f1 is both necessary and sufficient for the intrinsic neural conversion of ESCs.

## Pou3f1 promotes the neural transition from epiblast to neural progenitor cells

Our previous study showed that ESC neural differentiation could be divided into two stages: ESCs to EpiSCs and EpiSCs to NPCs (*Zhang et al., 2010a*). Therefore, we investigated which stage of neural differentiation is regulated by Pou3f1. To address this question, we performed ESC-derived EpiSC (ESD-EpiSC) colony formation assays (*Zhang et al., 2010a*) using day 2 ESC aggregates in serum-free medium. The results demonstrated that both control and Pou3f1-overexpressing ESCs generated similar numbers of homogeneous compact monolayer EpiSC-like colonies that displayed weak alkaline phosphatase activity (AKP) and similar levels of Oct4 expression (*Figure 2A,B*). Furthermore, both types of EpiSC-like colonies expressed comparably high levels of the pluripotency markers *Oct4* and *Nanog*, and of the epiblast marker *Fgf5*, with the absence of the expression of the ESC-specific gene *Rex1* (*Figure 2D*). Consistently, Pou3f1 knockdown did not affect the formation and markers' expression of EpiSC-like colonies (*Figure 2C,E*). These results suggest that Pou3f1 may not be involved in the first stage of ESC neural differentiation.

To determine whether Pou3f1 plays a role at the second stage of ESC neural differentiation, we used Dox to induce Pou3f1 overexpression during various periods in the ESD–EpiSC colony formation assay. The short-term overexpression of Pou3f1 was achieved by adding Dox for the first 2 days (+Dox 0–2), whereas the long-term overexpression was achieved by adding Dox for 6 days (+Dox 0–6) (*Figure 2F*). The number and morphology of EpiSC-like colonies from the short-term treated ESCs were similar to those characteristics of untreated control ESCs (−Dox). Additionally, AKP and Oct4 expression levels were also similar to those levels in the controls (*Figure 2F,G*). However, the number of EpiSC-like colonies from the long-term treated ESCs was significantly reduced, as was the expression of AKP and Oct4, whereas the expression of neural makers, such as Nestin and Tuj1, increased (*Figure 2F,G*). Moreover, the enhanced expression of *Sox1*, *Pax6*, and *Nestin* was also confirmed by Q-PCR (*Figure 2H*). Therefore, these results suggest that Pou3f1 may function during the second stage of ESC neural differentiation, from EpiSCs to NPCs.

To validate this finding, EpiSCs derived from early mouse embryos were differentiated in serum-free medium for 4 days. Gene expression profiling revealed that *Pou3f1* transcripts peaked at

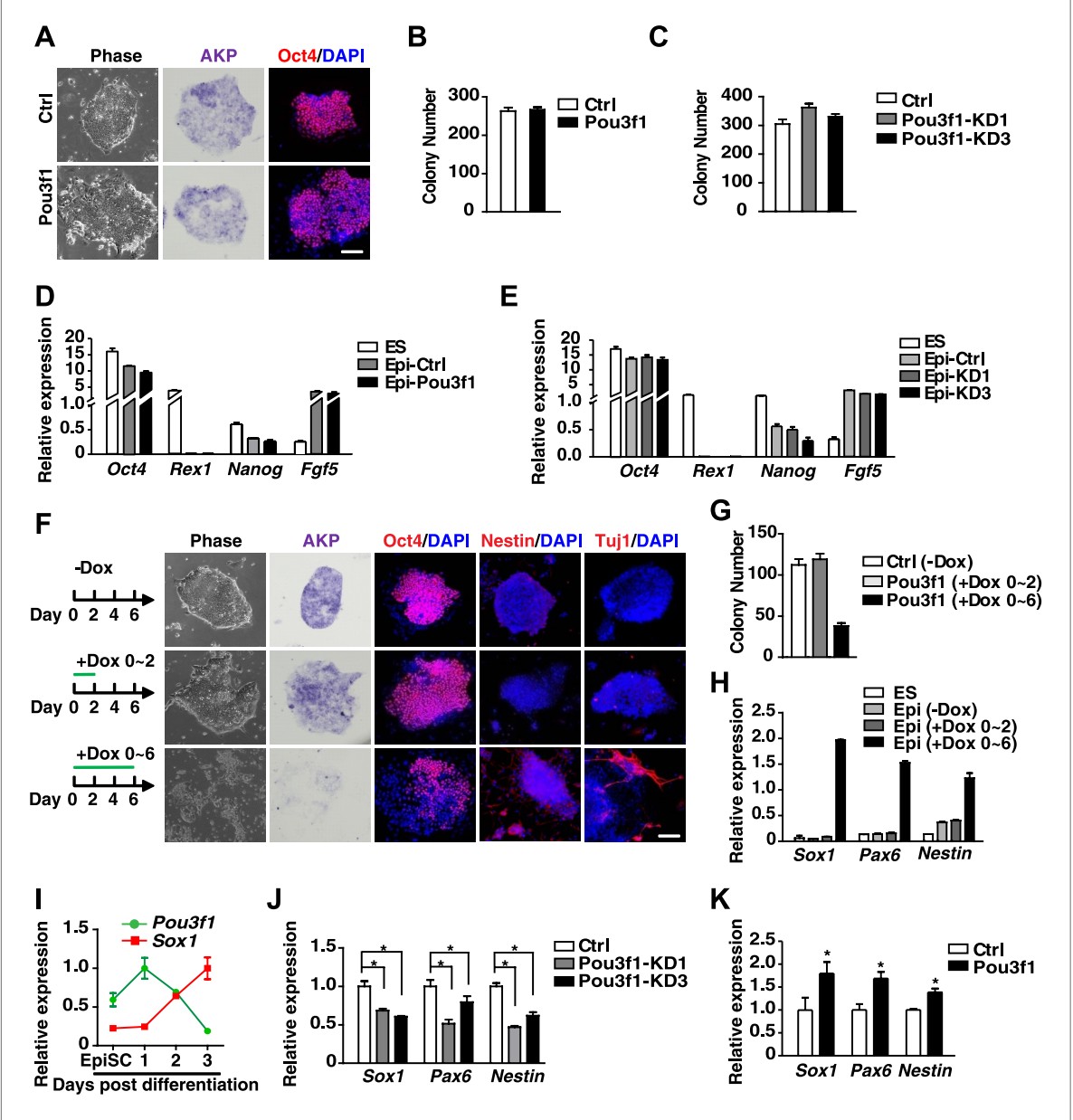

**Figure 2**. Pou3f1 promotes the neural differentiation from EpiSCs to NPCs. (**A**) Inducible Pou3f1-overexpressing ESCs were cultured as EBs for 2 days in the medium with or without Dox and then subjected to the ESD-EpiSC colony formation assay for 6 days in Dox-free CDM/AF medium. EpiSC-like colony cellular morphology, alkaline phosphatase activity (AKP) (purple), and Oct4 immunostaining (red) are presented. Scale bars, 100 μm. (**B**) Statistical analysis of EpiSC-like colonies in **A**. (**C**) Statistical analysis of EpiSC-like colonies from the control-ESCs and from Pou3f1-knockdown ESCs (Pou3f1-KD1, Pou3f1-KD3) in the ESD-EpiSC colony formation assay. (**D**) Gene expression levels in ESCs and in EpiSC-like colonies formed in **A**. (**E**) Gene expression levels in ESCs and in EpiSC-like colonies formed in **C**. (**F**) EpiSC-like colonies from control-ESCs (−Dox), short-term Pou3f1-overexpressing ESCs (+Dox 0–2), and from long-term Pou3f1-overexpressing ESCs (+Dox 0–6) in the ESD-EpiSC colony formation assay. Cellular morphology, AKP activity, and immunostaining for Oct4, Nestin, or for Tuj1 with DAPI are presented. Scale bars, 100 μm. (**G**) Statistical analysis of EpiSC-like colony numbers described in **F**. (**H**) Gene expression levels of ESCs and of the EpiSC-like colonies described in **F**. (**I**) Expression profiling of *Pou3f1* and *Sox1* during EpiSC neural differentiation in serum-free medium. (**J**) Gene expression levels of control and Pou3f1-knockdown EpiSCs in serum-free medium at differentiation day 2 were determined by Q-PCR. (**K**) Gene expression levels of control and Pou3f1-overexpressing EpiSCs at unbiased EBs differentiation day 2 were determined by Q-PCR. The values represent the mean ± SD for **B**–**E** and for **G**–**K**. (*$p<0.05$).

differentiation day 1 and subsequently declined with the onset of *Sox1* expression (*Figure 2I*). In Pou3f1-knockdown EpiSCs at neural differentiation day 2, *Sox1*, *Pax6*, and *Nestin* expression was reduced (*Figure 2J*), whereas *Sox1*, *Pax6*, and *Nestin* expression was increased in Pou3f1-overexpressing

EpiSCs at unbiased differentiation day 2 (*Figure 2K*). These results suggest that Pou3f1 facilitates the neural differentiation of EpiSCs. Together, these data indicate that Pou3f1 promotes pluripotent stem cell neural differentiation during the transition from EpiSCs to NPCs.

## Pou3f1 promotes the neural fate commitment of pluripotent stem cells in chimeric mouse embryos

To explore the function of Pou3f1 in vivo, first, we verified *Pou3f1* expression patterns in early mouse embryos by in situ hybridization. *Pou3f1* transcripts were detected in the whole epiblast and in the extraembryonic region of mouse embryos at E5.5 (*Figures 3A*, a, g). Then, *Pou3f1* expression was gradually restricted to the anterior part of the embryos from E6.5 to E7.0 (*Figure 3A*, c, d). Transverse sections of embryos revealed that *Pou3f1* expression was exclusively localized to the anterior region of the inner epiblast, which would prospectively undergo neuroectoderm fate (*Figures 3A*, b–d, h, i). During the neural initiation stage at E7.5 and at E8.0, *Pou3f1* expression was further restricted to the anterior neuroectoderm (*Figure 3A*, e, f, j, k), suggesting a causal correlation with embryonic neural differentiation.

Next, we performed a blastocyst injection study using manipulated ESCs. GFP-labeled control, Pou3f1-knockdown (Pou3f1-KD), and Pou3f1-overexpressing (Pou3f1-OE) ESCs were injected into E2.5 blastocysts and transferred into pseudopregnant mice, respectively. The developmental potentials of these cells were examined after 7 days post-transplantation (at E9.0–E9.5). Chimeras were generated from these three ES cell lines (*Figure 3—figure supplement 1*). The number of GFP-positive cells in various tissues was ascertained in sections of chimeric embryos. The control ESCs contributed to a wide range of germ layer lineages, including neuroectoderm (NE), mesenchyme (M), somite (S), heart, gut, and extraembryonic ectoderm, at similar percentages (~60%) (*Figure 3B,C*). Surprisingly, only Pou3f1-KD ESCs failed to contribute to the neuroectoderm, but were widely identified in non-neural lineages (*Figure 3B,C*). By contrast, Pou3f1-OE ESCs preferred incorporation into the neuroectoderm and displayed a considerably reduced contribution to non-neural tissues (*Figure 3B,C*). These results indicate that Pou3f1 promotes the neural fate commitment of pluripotent stem cells in vivo.

## Genome-wide ChIP-seq and RNA-seq analyses of Pou3f1

To investigate the regulatory mechanism of Pou3f1 at the global level, we performed RNA-seq assays to identify Pou3f1-regulated genes during ESC differentiation. Pou3f1-overexpressing ESCs were differentiated in unbiased medium, and total RNAs were collected from EBs at days 2, 4, and 6 for mRNA sequencing. The RNA-seq analysis revealed that the global transcriptome changed dramatically from day 2 to day 6 (*Figure 4A*). Because day 4 EBs were at the transition state from the epiblast-like stage at day 2 to the NPC-like stage at day 6 (*Zhang et al., 2010a*), we focused on the transcriptome data from day 4. To validate the deep-sequencing data, we examined the expression levels of approximately 30 genes by Q-PCR and found that these expression levels were consistent with the sequencing data (*Figure 4—figure supplement 1A*). Of the 11,356 genes expressed (rpkm > 1), 768 genes were up-regulated, and 202 genes were down-regulated (Cuffdiff, FDR < 0.05).

To identify genes directly regulated by Pou3f1 on a genome-wide scale, ChIP-seq assays were performed with day 4 EBs. Interestingly, a large percentage of Pou3f1-binding sites (47%) were located in distal regions more than 50 kb away from known or predicted transcription start sites (TSS). Only a small percentage of Pou3f1 binding sites resided in 5′ proximal regions (0–1 kb and 1–5 kb), reflecting the property of Pou3f1 to control transcription primarily through distal enhancers. To investigate whether Pou3f1 binding to the genomic regions exerts functional consequences through regulating targeted gene expression, we integrated the ChIP-seq data with the RNA-seq data. Among the 4674 Pou3f1-binding genes, 430 genes were modulated significantly (*Figure 4C*). Gene Ontology term enrichment analysis revealed that genes up-regulated by Pou3f1 were primarily involved in neural differentiation processes, such as neuron differentiation, neuron development, and axonogenesis, whereas Pou3f1-down-regulated targets were highly enriched in pattern specification and in embryonic morphogenesis (*Figure 4D*).

Detailed ChIP-seq and RNA-seq analyses showed that the genomic region of neural development-related genes, such as *Pax6*, *Sox2*, and *Zfp521*, was bound by Pou3f1 and that their expression was up-regulated by Pou3f1 overexpression. Intriguingly, the downstream targets of important morphogens, such as *Gata4* in the BMP pathway as well as *Myc* and *Dkk1* in the Wnt pathway, were also bound by Pou3f1. However, the expression of these genes was down-regulated by Pou3f1 overexpression (*Figure 4E*). Pou3f1 genomic binding was confirmed by ChIP-qPCR (*Figure 4—figure supplement 1B*). We also found that Pou3f1 could bind to the genomic regions of *Zic1* and of *Zic2*, which are related to

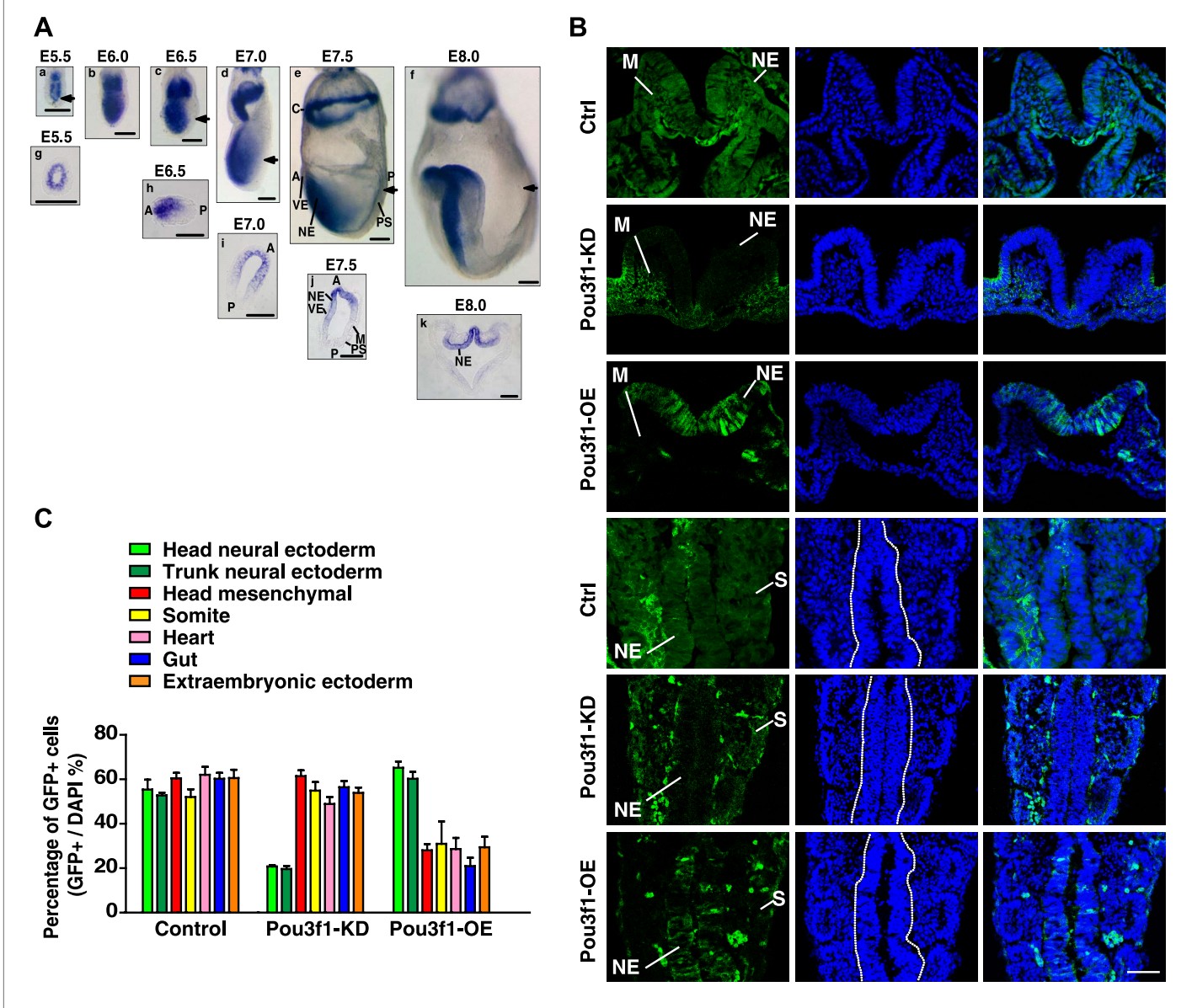

**Figure 3**. Pou3f1 promotes neural fate commitment in vivo. (**A**) Whole-mount in situ hybridization of *Pou3f1* in early mouse embryos (E5.5–E8.0). The arrowhead marks the position-plane of the transverse section of the corresponding embryo below. Scale bars, 100 μm. (**B**) Contribution of injected GFP-labeled control (Ctrl), Pou3f1-knockdown (Pou3f1-KD), and inducible Pou3f1-overexpressing (Pou3f1-OE) ESCs to different germ lineages in chimeric embryos. NE, neuroectoderm; M, mesenchyme; and S, somite. Scale bars, 50 μm. (**C**) Statistical analysis of GFP-positive cell distribution in the various germ layer lineages in the ESC blastocyst injection study. The values represent the mean ± SD for **C**.

The following figure supplements are available for figure 3:

**Figure supplement 1**. Information of chimeric mice generated from *Pou3f1*-overexpressing or knockdown ESCs.

neural development, and of the BMP and Wnt signaling targets *Id1* and *Axin2* (**Figure 4—figure supplement 1C**). Together, these results suggest that Pou3f1 might promote ESC neural fate commitment through regulating the expression of multiple genes.

## Pou3f1 increases neural development-related gene expression

Genome-wide ChIP-seq and RNA-seq assays revealed that Pou3f1 might regulate a group of genes related to neural development, such as *Sox2*, *Zfp521*, *Zic1*, and *Zic2* (**Figure 4E**, **Figure 4—figure supplement 1C**). Q-PCR confirmed that expression of these neural fate-promoting factors was

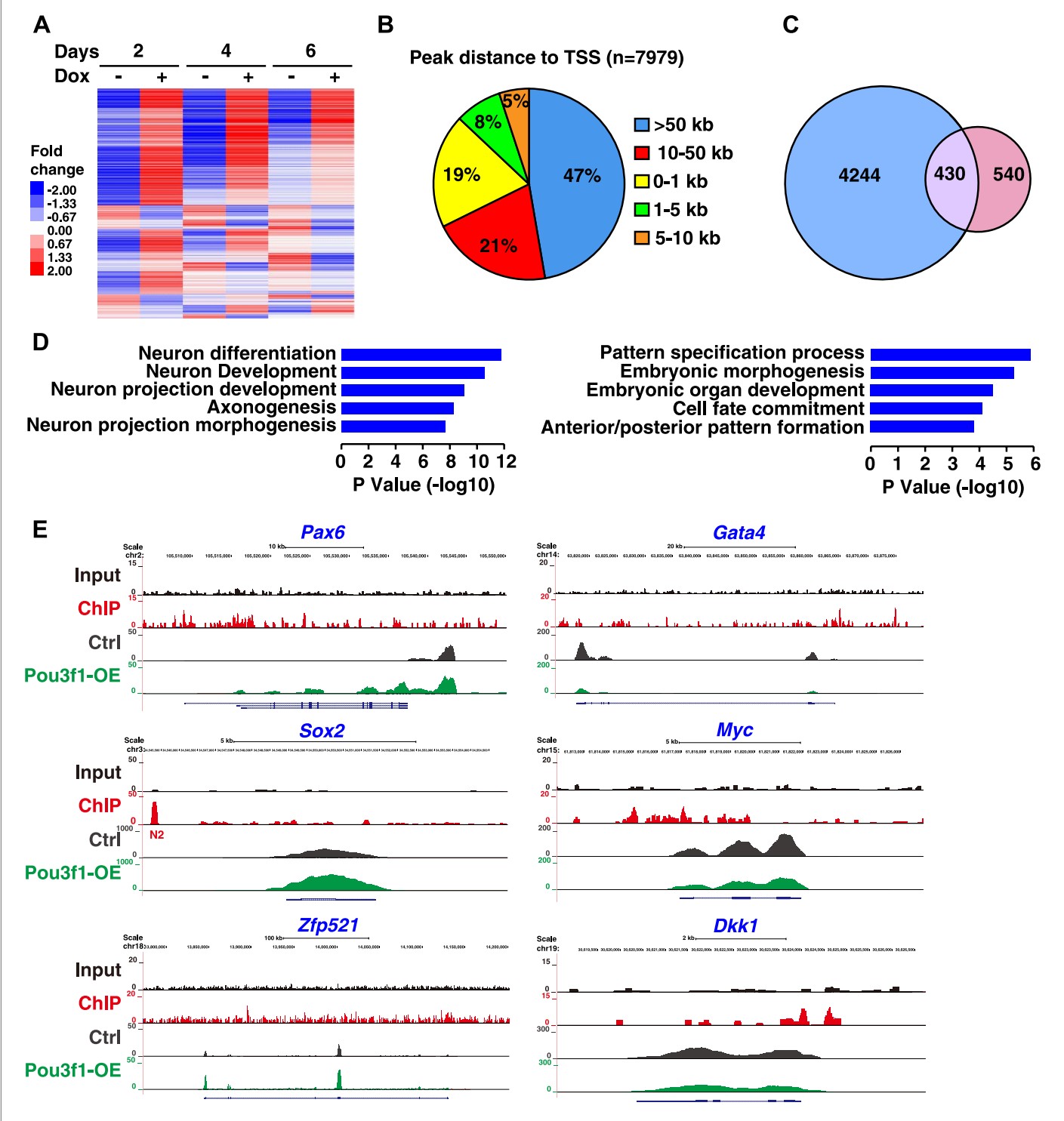

**Figure 4**. RNA-seq and ChIP-seq analysis of Pou3f1 downstream targets. (**A**) RNA-seq gene expression heat map of control and of inducible Pou3f1-overexpressing ESCs with Dox-treatment for 6 days. Heat-map colors (red, up-regulation; blue, down-regulation) indicate gene expression in units of standard deviation from the mean of all samples. (**B**) Analysis of Pou3f1-enriched regions in the ChIP-seq assay. Pie chart showing the percentage distribution of Pou3f1-binding peaks in each category. The ChIP-seq assay was performed with Pou3f1-overexpressing ESCs at differentiation day 4. (**C**) Venn diagram depicting the overlap (purple) of Pou3f1-bound genes (blue) and genes with significantly altered expression upon *Pou3f1* overexpression (pink) at differentiation day 4. Statistical significance was estimated by Fisher's exact test (p<4.71e−75). (**D**) GO analysis of biological processes of the overlap genes described in **C**. Many genes involved in neural differentiation processes were up-regulated, whereas a few genes related to pattern

*Figure 4. Continued on next page*

*Figure 4. Continued*

specification were down-regulated. Log p value was used to rank the enrichment. (**E**) Genome browser view of the distribution of the ChIP-seq and RNA-seq reads of represented genes. The upper panels show the Pou3f1-binding regions identified by ChIP-seq (black, input; red, Pou3f1-binding site at genomic loci), and the lower panels depict the RNA-seq reads of the represented genes in control ESCs (gray) and in Pou3f1-overexpressing ESCs (green) at differentiation day 4.

The following figure supplements are available for figure 4:

**Figure supplement 1**. Pou3f1 is enriched in the loci of multiple downstream target genes.

decreased by Pou3f1 knockdown and increased by Pou3f1 overexpression (*Figure 5A,B*). Next, we investigated how *Pou3f1* regulates the expression of these target genes. As a transcription factor, Pou3f1 contains three domains: the amino-terminal region, the POU domain, and the HOMEO domain. The POU domain and HOMEO domains mediate protein interactions and DNA binding (*Levavasseur et al., 1998*). Among serial deletion mutants (*Figure 5—figure supplement 1A*), the HOMEO domain deleted mutant (ΔHOMEO) exclusively failed to promote ESC neural differentiation (*Figure 5—figure supplement 1B* and data not shown). This result suggests that the HOMEO domain is essential for the Pou3f1-mediated promotion of the neural fate.

The expression of *Sox2*, which is an important neural induction gene, is regulated by different enhancers. For example, the N2 enhancer regulates *Sox2* expression in the anterior neural plate, and the N1 enhancer regulates *Sox2* expression in the posterior neural plate (*Uchikawa et al., 2003*; *Takemoto et al., 2011*). Our ChIP-seq data revealed that Pou3f1 binds to the N2 enhancer region of the *Sox2* gene and promotes *Sox2* expression (*Figure 4E*). To further confirm this regulation, luciferase assays were conducted using a reporter construct driven by the *Sox2*N2 enhancer. Wild-type Pou3f1 enhanced luciferase activity; however, the ΔHOMEO mutant did not enhance this activity (*Figure 5C*). Similarly, ChIP assays revealed that wild-type Pou3f1, but not Pou3f1-ΔHOMEO, bound to the *Sox2*N2 enhancer, and neither of them bound to the *Sox2*N1 enhancer (*Figure 5D*). Thus, Pou3f1 regulates *Sox2* expression by binding to the N2 enhancer, and this activity is mediated by the HOMEO domain.

Chick embryos have been widely used as an in vivo model to study early neural development (*Stern, 2005a*). In early chick embryos, chick *Pou3f1* (*cPou3f1*) was initially expressed at the anterior portion of the primitive streak at HH stage 3+ (*Figure 5E*, a). Then, the territory of cPou3f1 expanded to the prospective neural plate, where the earliest expression of *cSox2* was detectable at HH stage 4 (*Figure 5E*, b, i, j). From HH stage 5 onward, *cPou3f1* expression highly overlapped with *cSox2* in the anterior neural plate (*Figure 5E*, c–h, k–p). These results demonstrated that *cPou3f1* was expressed earlier than *cSox2* in the prospective neural plate in early chick embryos, suggesting that *cPou3f1* activates *cSox2* expression in chick embryos. These results are similar to our findings concerning ESC neural differentiation.

To determine the function of Pou3f1 in early chick embryos, either the control vector or Pou3f1 was electroporated into the epiblast layer of HH stage 3 chick embryos as a line extending outwards from the prospective neural plate (*Linker et al., 2009*), and the expression of *cSox2* was analyzed 12 hr later. The ectopic expression of Pou3f1 induced the lateral expansion of *cSox2* expression (7/9), whereas the control vector did not (0/9) (*Figure 5F*). Taken together, these results suggest that Pou3f1 promotes neural fate commitment by directly activating the expression of neural development-related genes.

## Pou3f1 inhibits the BMP and Wnt pathways by interfering with their transcriptional activities

In addition to the direct regulation of intrinsic factors, Pou3f1 might also interfere with the activities of extrinsic inhibitory signals, such as the BMP and Wnt pathways, in ESC neural differentiation (*Figure 4E*, *Figure 4—figure supplement 1C*). Indeed, during ESC neural differentiation, Pou3f1 knockdown increased the expression of the BMP targets *Id1*, *Id2*, *Msx1*, and *Msx2* (*Figure 6A*), whereas Pou3f1 overexpression generated the opposite effect (*Figure 6B*). In vivo electroporation studies also revealed that the ectopic expression of Pou3f1 reduced the expression of the BMP target gene *cId1* (6/10) at the edge of the chick anterior peripheral ectoderm, whereas the control vector did not (0/11) (*Figure 6—figure supplement 1A*).

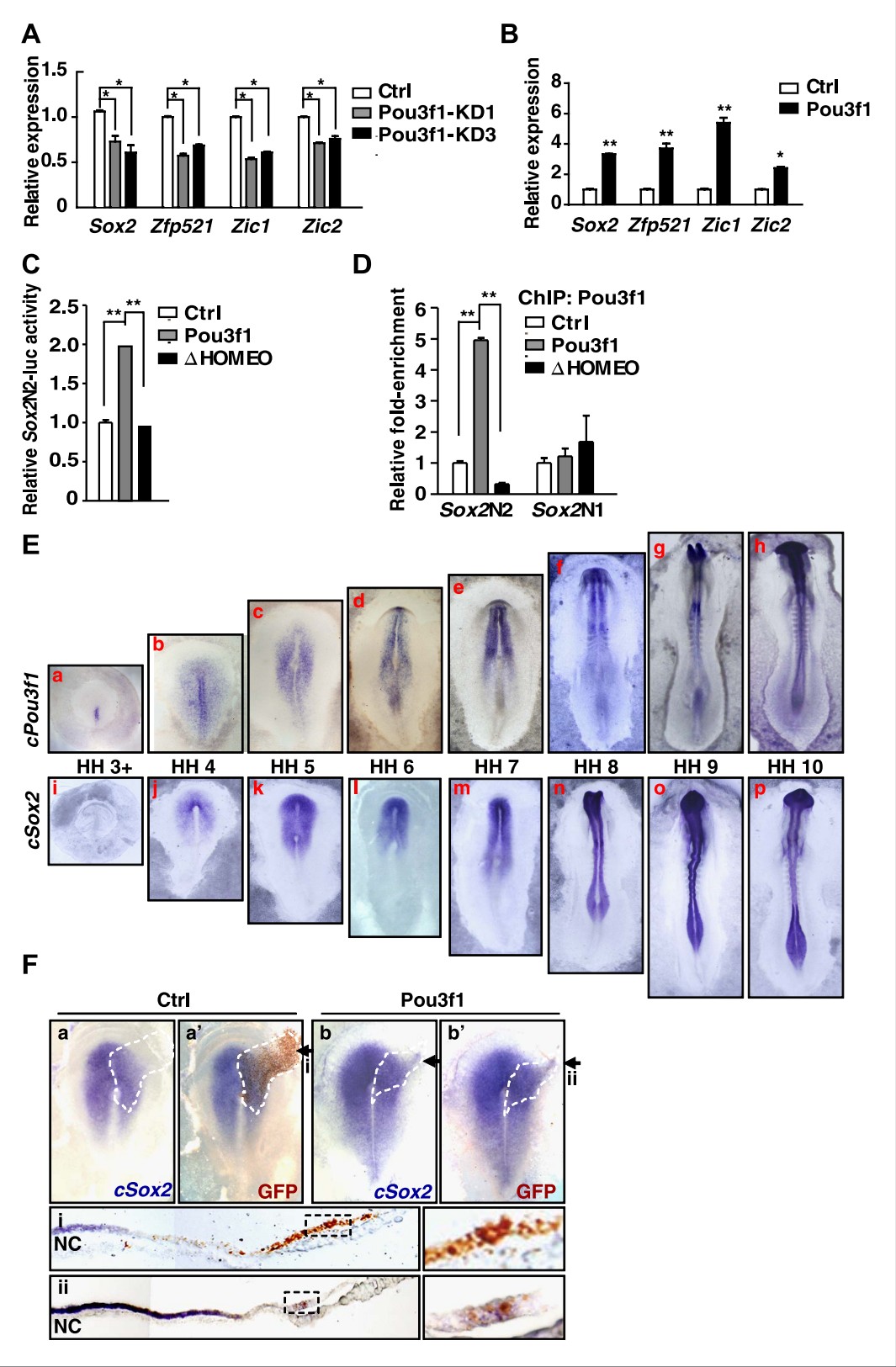

**Figure 5**. Pou3f1 increases neural lineage-specifier expression. (**A**) Gene expression levels in control and in Pou3f1-knockdown ESCs differentiated in serum-free medium for 4 days. (**B**) Gene expression levels in control and

*Figure 5. Continued on next page*

*Figure 5. Continued*

in inducible Pou3f1-overexpressing ESCs at unbiased differentiation day 8. (**C**) Luciferase assays using the *Sox2*N2-luc enhancer in control, Pou3f1-full length, or in Pou3f1-ΔHOMEO vector-transfected HEK293 cells. (**D**) ChIP assay in control, Pou3f1-full length, or in Pou3f1-ΔHOMEO lentivirus-transfected P19 cells. A Pou3f1-specific antibody was used, and Pou3f1 enrichment at *Sox2*N2 and *Sox2*N1 enhancer regions was normalized to the *Sox2* coding region. (**E**) Whole-mount in situ hybridization of *cPou3f1* (a–h) and *cSox2* (i–p) in early chick embryos from HH stage 3+ to HH stage 10. (**F**) Pou3f1 overexpression induces *cSox2* expression ectopically. IRES-GFP (control vector, a and a') or Pou3f1-IRES-GFP (b and b') was electroporated into the epiblast layer of the chick embryos. *cSox2* (blue) expression was examined by in situ hybridization (a, b, a', b'). GFP expression (brown) indicating the electroporated field was detected by immunohistochemical assays (a' and b'). The arrowhead marks the position-plane of the corresponding embryo transverse section below (i and ii). NC, notochord. The values represent the mean ± SD for **A**–**D**. (*p<0.05; **p<0.01).

The following figure supplements are available for figure 5:

**Figure supplement 1**. HOMEO domain is essential for the neural-promoting effect of Pou3f1.

Then, we explored the mechanism underlying the Pou3f1-mediated inhibition of BMP targets. Luciferase assays were performed with a four-repeat BMP responsive element (BRE)-driven reporter (*Katagiri et al., 2002*) to examine BMP activity in ESCs and in P19 cells. Pou3f1 knockdown increased BRE activity in ESCs with or without BMP4 stimulation (*Figure 6C*), whereas Pou3f1 overexpression partially inhibited BRE-luc activity (*Figure 6D*). Furthermore, among the several known functional domains, only the HOMEO domain was necessary to maintain the inhibitory effect of Pou3f1 on BMP signaling (*Figure 6—figure supplement 1B*). ChIP assays using a Pou3f1 antibody were performed, and we found that the binding of wild-type Pou3f1, rather than Pou3f1-ΔHOMEO, was specifically enriched at the BRE region of the *Id1* gene promoter (*Figure 6E*). We also performed ChIP assays using a pSmad1 antibody and found that pSmad1 bound to the BRE locus of the *Id1* promoter, but not to the 3′ UTR region, in the presence of BMP4 (*Figure 6F*, open column). Interestingly, Pou3f1 interfered with the binding of pSmad1 to the BRE locus of the *Id1* promoter (*Figure 6F*, filled column). Moreover, Pou3f1 repressed BMP-induced luciferase activity in a dose-dependent manner (*Figure 6G*). We also observed that Pou3f1 did not affect the stimulation, degradation, dephosphorylation, or intracellular translocation of pSmad1 (data not shown), excluding the fact that Pou3f1 regulates the BMP pathway through a signaling cascade. Together, these results suggest that Pou3f1 may inhibit BMP signaling by interfering with pSmad1 binding to the regulatory elements and then repressing the transcription of target genes.

Similar to the BMP pathway, *Wnt3a*, *Axin2*, *Dkk1*, and *Myc* in Wnt signaling were regulated by Pou3f1 during ESC neural fate commitment (*Figure 6H,I*). ChIP-seq data revealed that Pou3f1 directly binds to the promoter regions of these Wnt signaling targets (*Figure 4E*, *Figure 4—figure supplement 1C*). In luciferase-based TOPflash (TCF optimal promoter) Wnt reporter assays (*Korinek et al., 1997*), TOPflash-luc activity was enhanced by Pou3f1 knockdown (*Figure 6J*), and Wnt3a-induced luciferase activity was partially reduced with Pou3f1 overexpression (*Figure 6K*). We also found that the HOMEO domain is crucial for sustaining the inhibitory effect of Pou3f1 on TOPflash-luc activity (*Figure 6—figure supplement 1C*). Pou3f1 also inhibited Wnt agonist CHIR99021-induced TOPflash-luc activity in a dose-dependent manner (*Figure 6L*). Together, these results suggest that Pou3f1 interferes with the BMP and Wnt signaling pathways by directly inhibiting the transcription of their target genes.

## Pou3f1 rescues the neural inhibition effects of the BMP and Wnt pathways

The BMP and Wnt signaling pathways have strong inhibitory effects on ESC neural differentiation (*Haegele et al., 2003*; *Ying et al., 2003*), and the above data suggest that Pou3f1 inhibits BMP and Wnt transcriptional activities. Thus, we investigated whether Pou3f1 could attenuate their inhibitory effects. ESCs were differentiated in serum-free medium with or without BMP4 for 48 hr from day 2 to day 4, and Dox was simultaneously added to induce Pou3f1 overexpression (*Figure 7A*). Consistent with our previous observation (*Zhang et al., 2010a*), BMP4 inhibited the expression of the neural markers Sox1, Pax6, Nestin, and Tuj1 at both the mRNA and protein levels (Ctrl BMP4$^+$ compared with Ctrl

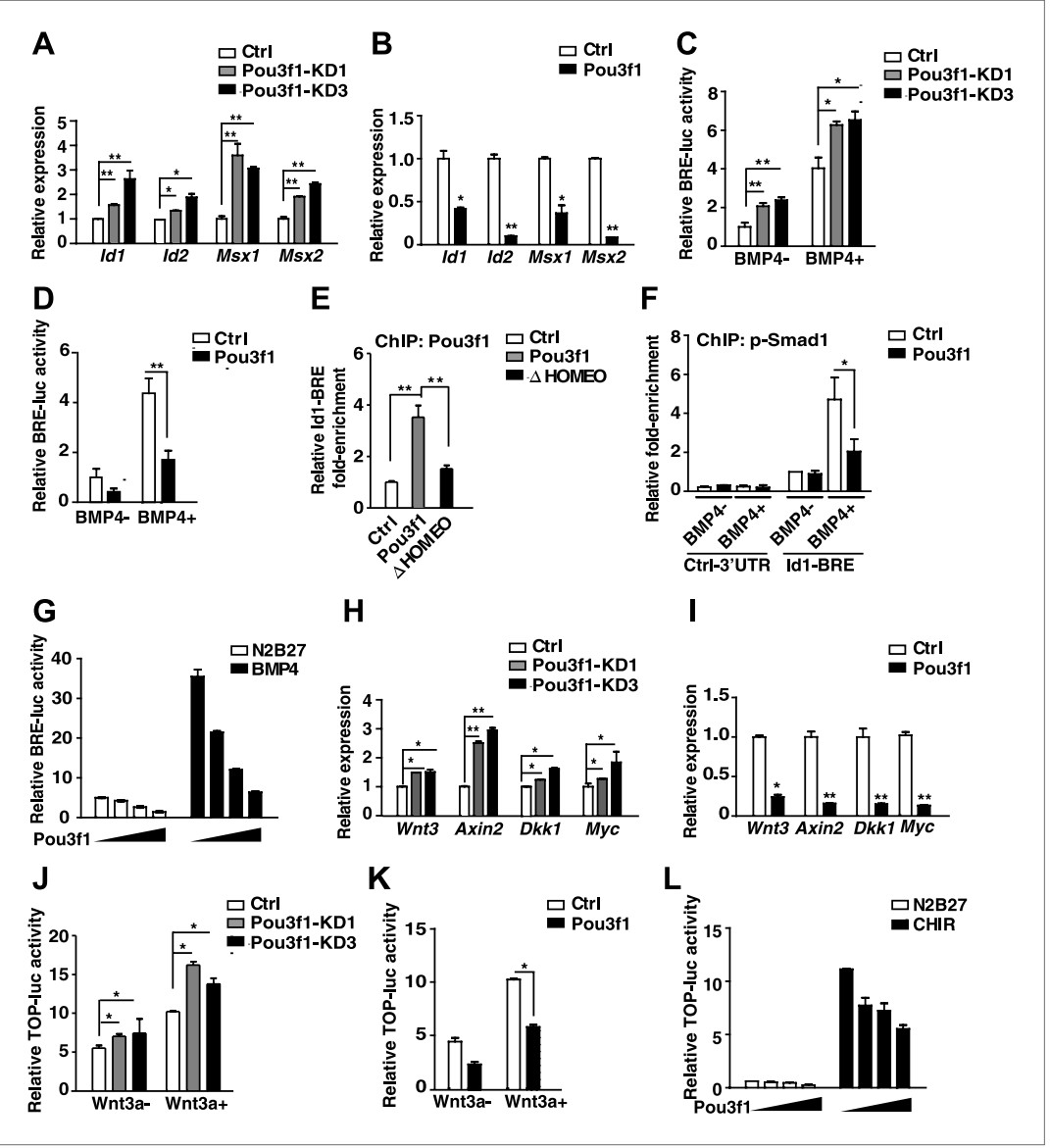

**Figure 6**. Pou3f1 represses BMP and Wnt signaling at the transcriptional level. (**A**) Expression levels of BMP signaling target genes in control and Pou3f1-knockdown ESCs differentiated in serum-free medium. (**B**) Expression levels of BMP signaling target genes in control and Pou3f1-overexpressing ESCs in unbiased differentiation. (**C**) Luciferase assays using BRE-luc in control and Pou3f1-shRNA vector-transfected ESCs with or without BMP4 treatment in N2B27 medium. (**D**) Luciferase assays using BRE-luc in control and Pou3f1-expressing vector-transfected ESCs with or without BMP4 treatment in N2B27 medium. (**E**) Pou3f1 ChIP assays in control, Pou3f1-full length, or in Pou3f1-ΔHOMEO lentivirus-transfected P19 cells. Pou3f1 enrichment at the *Id1*-BRE was normalized to the *Id1* 3′ UTR region. (**F**) pSmad1 ChIP assay in control and Pou3f1-full length lentivirus-transfected P19 cells with or without BMP4 treatment. A pSmad1/5/8-specific antibody was used in the assay. pSmad1 enrichment at the *Id1*-BRE and control 3′ UTR region were analyzed. (**G**) Dose-dependent inhibitory effect of Pou3f1 on the BRE-luc reporter activities. P19 cells were transfected with increasing amounts of Pou3f1-expressing vector and treated with or without BMP4 in N2B27 medium. (**H**) Expression levels of Wnt signaling target genes in control and Pou3f1-knockdown ESCs differentiated in serum-free medium. (**I**) Expression levels of Wnt signaling target genes in control and Pou3f1-overexpressing ESCs in unbiased differentiation. (**J**) Luciferase assays using TOPflash in control and Pou3f1-shRNA vector-transfected ESCs with or without stimulation of Wnt3a in N2B27 medium. (**K**) Luciferase assays using TOPflash in control and Pou3f1-expressing vector-transfected ESCs with or without stimulation of Wnt3a in N2B27 medium. (**L**) Dose-dependent inhibitory effect of Pou3f1 on the TOPflash luciferase reporter

*Figure 6. Continued on next page*

*Figure 6. Continued*

activities. P19 cells were transfected with increasing amounts of Pou3f1-expressing vector and treated with or without CHIR99021 in N2B27 medium. The values represent the mean ± SD. (*p<0.05; **p<0.01).
The following figure supplements are available for figure 6:

**Figure supplement 1**. Pou3f1 interferes with BMP and Wnt signaling pathways at the transcriptional level.

BMP4⁻ in *Figure 7A–C*). As expected, Pou3f1 overexpression fully restored the expression of these markers in ESC neural differentiation (Pou3f1 BMP4⁺ compared with Ctrl BMP4⁺ in *Figure 7A–C*). Furthermore, Pou3f1 overexpression also fully rescued the neural inhibitory effects of Wnt3a (*Figure 7D*). To test whether Pou3f1 relieves the neural inhibition mediated by the BMP signaling pathway in vivo,

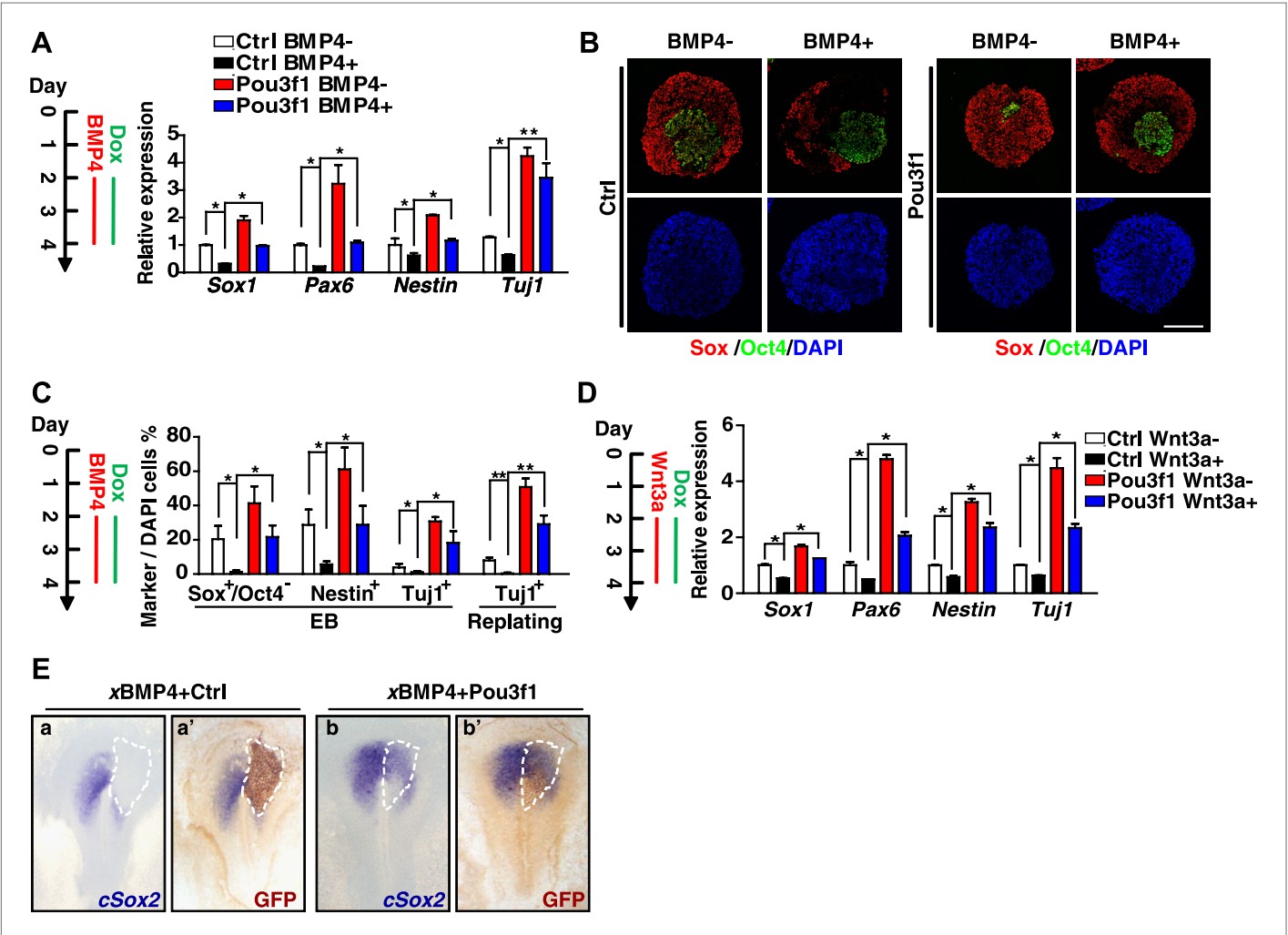

**Figure 7**. Pou3f1 alleviates the inhibitory effects of BMP4 and Wnt3a on neural fate commitment. (**A**) Inducible Pou3f1-overexpressing ESCs were cultured as EBs in serum-free medium for 4 days with or without BMP4/Dox treatment from day 2 to day 4. Gene expression levels were detected by Q-PCR. (**B**) Immunocytochemical assays using day 4 EBs described in **A**. The EBs were stained with Sox (red) and with Oct4 (green). Scale bars, 100 μm. (**C**) Statistical analysis of results from the immunocytochemical assay of Sox⁺/Oct4⁻, Nestin⁺, and Tuj1⁺ cells in EBs and of Tuj1⁺ replated cells. (**D**) Pou3f1-overexpressing ESCs were cultured as EBs in serum-free medium for 4 days with or without Wnt3a/Dox addition from day 2 to day 4. Gene expression levels were detected by Q-PCR. (**E**) *Pou3f1* partially rescues the inhibitory effects of *x*BMP4 on *cSox2*. In situ hybridization of *cSox2* (blue) in chick embryos that were co-electroporated with *x*BMP4 plus IRES-GFP (control vector, a and a') or Pou3f1-IRES-GFP (b and b'), respectively. GFP expression (brown) was detected in a' and b' by immunohistochemistry. The values represent the mean ± SD for **A**, **C**, and **D**. (*p<0.05; **p<0.01).

we co-electroporated *Xenopus* BMP4 (*xBMP4*) with a control vector or with Pou3f1 into the pre-neural plate region of chick embryos at HH stage 3. *cSox2* expression was completely repressed by *xBMP4* (16/19), whereas the forced expression of Pou3f1 partially recovered *cSox2* expression (10/19) (*Figure 7E*). Together, these results suggest that Pou3f1 alleviates the inhibitory activities of both BMP and Wnt signals during neural fate commitment.

## Discussion

In the past decades, studies on early neural development have mainly focused on the role of extrinsic signals. Recent works have provided new insights concerning the intracellular programs involved in early neural fate commitment in the absence of extracellular signals (*Kamiya et al., 2011*; *Iwafuchi-Doi et al., 2012*). However, how the intrinsic and extrinsic regulatory networks are orchestrated to ensure the appropriate initiation of neural differentiation remains largely unclear. Our in vitro and in vivo data indicate that Pou3f1 is crucial for ESC neural fate commitment and promotes the transition from EpiSCs to neural progenitor cells. Furthermore, Pou3f1 functions as an intrinsic regulator of both intracellular transcription factors and extracellular inhibitory signals during neural fate commitment.

Pou3f1 was previously reported to be a transcription factor that participates in Schwann cell development and myelination (*Bermingham et al., 1996*; *Jaegle et al., 1996*). The *Pou3f1* gene expression profiles in mouse embryos in vivo (*Figure 3*; *Zwart et al., 1996*) and of ESC differentiation in vitro (*Figure 1*, *Figure 1—figure supplement 1*) imply that Pou3f1 may also participate in early neural development. Indeed, the shRNA-mediated knockdown of *Pou3f1* in ESCs results in the reduced expression of the neural markers *Sox1*, *Pax6*, and *Tuj1* in serum-free medium (*Figure 1*). However, the compensation of the POU III member Brn2 may be one of the reasons for the mild effects observed during ESC neural differentiation after *Pou3f1* depletion (*Figure 1*, *Figure 1—figure supplement 2*). Brn2 compensation and the different ESC lines and culture system used potentially explain why the Pou3f1 knockdown effects are not reported in Iwafuchi-Doi's study (*Iwafuchi-Doi et al., 2012*). On the other hand, our results are consistent with their results indicating that the forced expression of *Pou3f1* promotes the expression of neural markers (*Figure 1*, *Iwafuchi-Doi et al., 2012*). Clearly, Pou3f1 is necessary and sufficient for ESC neural differentiation. Pou3f1-overexpressing or Pou3f1-knockdown ESCs generate EpiSC-like colonies that are similar to the control ESCs. However, the neural differentiation of Pou3f1-overexpressing or Pou3f1-knockdown EpiSCs is markedly different from the control EpiSCs, suggesting that Pou3f1 functions specifically during the neural transition from the epiblast to neural progenitor cells (*Figure 2*). Furthermore, in our blastocyst injection study, the contribution of Pou3f1-knockdown ESCs to the neuroectoderm was severely impaired (*Figure 3*), indicating that Pou3f1 most likely functions cell-autonomously during the neural fate commitment of pluripotent stem cells in vivo. Our findings revealed that Pou3f1 is an essential transcription factor required for the intrinsic neural differentiation of pluripotent stem cells.

Cell fate determination is regulated in a step-wise fashion via the activation or inhibition of lineage specification factors (*Pfister et al., 2007*). Several transcription factors, including *Pax6*, *Sox2*, *Zfp521*, *Zic1*, and *Zic2*, promote neural gene expression and play roles in the derivation of the anterior neural plate (*Iwafuchi-Doi et al., 2012*; *Kamiya et al., 2011*; *Zhang et al., 2010b*). *Zfp521* and *Zic1/2* are important for neural fate consolidation rather than initiation (*Aruga, 2004*; *Kamiya et al., 2011*; *Iwafuchi-Doi et al., 2012*). To date, the intrinsic modulators essential for the early neural initiation event have not been identified. In this study, the combination of RNA-seq and ChIP-seq enabled us to investigate the underlying molecular mechanisms governing Pou3f1-mediated neural fate commitment in ESCs at the genome-wide level and to determine whether Pou3f1 is involved in the initiation of neural differentiation. Our results indicate that *Pax6*, *Sox2*, *Zfp521*, and dozens of other known neural fate-promoting genes are enhanced by Pou3f1 overexpression during ESC differentiation (*Figures 4 and 5*). Furthermore, ChIP-seq data reveal that Pou3f1 is enriched at the regulatory regions of *Pax6*, *Sox2*, *Zfp521*, *Zic1*, and *Zic2* genomic loci (*Figure 4*, *Figure 4—figure supplement 1*), indicating that *Pou3f1* directly activates these neural fate-promoting genes. Surprisingly, *Pou3f1* did not bind the *Sox2*N1 enhancer, which controls *Sox2* posterior neural plate expression; *Pou3f1* preferentially binds to the *Sox2*N2 enhancer, which drives *Sox2* anterior neural plate expression (*Figures 4 and 5*). This result is consistent with the in vivo *Pou3f1* and *Sox2* overlapping expression patterns during neural fate commitment. Our results are also consistent with the notion that the anterior-most portion of the epiblast constitutes the primitive neural identity following neural induction (*Andoniadou and Martinez-Barbera, 2013*; *Li et al., 2013*). Moreover, our observations confirm the hypothesis

proposed in a recent study (*Kamiya et al., 2011*) that *Pou3f1* functions upstream of *Zfp521* during ESC neural differentiation (*Figures 4 and 5*, *Figure 4—figure supplement 1*). Taken together, these findings demonstrate that Pou3f1 is most likely an intrinsic neural initiation factor that participates in the transition of pluripotent stem cells to NPCs by directly activating a group of key neural fate-promoting genes.

In addition to intrinsic factors, several extrinsic signals involved in early neural fate commitment have been intensively studied, including BMPs and Wnts. However, how BMP/Wnt inhibitory activities are alleviated to secure neural fate commitment has not been fully elucidated. BMP and Wnt signals function partially through their downstream genes (*ten Berge et al., 2011*; *Varlakhanova et al., 2010*; *Ying et al., 2003*; *Zhang et al., 2010a*). Unlike *Zfp521*, which did not affect BMP signaling (*Kamiya et al., 2011*), the expression of a few genes related to BMP and Wnt pathways was regulated by Pou3f1 knockdown or by overexpressing in EBs at day 4 (*Figure 6*). However, this regulation was not evident in ESCs or in EBs at day 2 (data not shown). This finding suggests that Pou3f1 interferes with the BMP/Wnt signaling pathways during the process of neural conversion from epiblast to NPCs. Moreover, Pou3f1 is recruited to the genomic loci of many downstream targets of BMP and Wnt signals, such as *Id1*, *Id2*, *Myc*, and *Axin2* (*Figure 4*, *Figure 4—figure supplement 1*). We also found that Pou3f1 represses the transcriptional activation of a BMP responsive element (BRE) by BMP4 and of a TCF optimal promoter (TOP) by Wnt3a (*Figure 6*, *Figure 6—figure supplement 1*). Our data further suggest that the binding of pSmad1 to the BRE locus is potentially compromised in the presence of Pou3f1, which results in the repression of BMP signaling pathway activity (*Figure 6*). However, other possibilities, such as the recruitment of repressing cofactors by Pou3f1, could not be excluded by the present study. Notably, Pou3f1 overexpression enables neural differentiation even in the presence of BMP4 or Wnt3a (*Figure 7*). We propose that the Pou3f1-dependent repression of the BMP and Wnt signaling pathways and the activation of intrinsic neural lineage genes together are involved in the neural fate-promoting activity of Pou3f1.

In summary, our study establishes Pou3f1 as a critical dual-regulator of intrinsic transcription factors and extrinsic signals to promote neural fate commitment. This study provides a better understanding of the internal mechanism of neural initiation. Nonetheless, many questions concerning this process remain unanswered, such as whether the dual regulatory mechanism of Pou3f1 is also utilized to initiate the mouse neural program in vivo, whether this two-way modulating processes occurs simultaneously or in a sequential, temporal manner, and how the controversial activation/inhibition activities of the Pou3f1 transcription factor is achieved. All these unanswered questions lay the foundation for exciting future work concerning the interplay between the extrinsic and intrinsic cues during early embryonic neural fate commitment.

## Materials and methods

### Cell culture, differentiation, and treatment

Mouse ESCs (R1 and R1/E) were maintained on feeders in standard medium. ESC serum-free neural differentiation (8% knockout serum replacement medium) and EB replating were performed as described previously (*Watanabe et al., 2005*; *Zhang et al., 2010a*). ESC unbiased differentiation in serum-containing medium (10% FBS) was performed as described previously (*Zhang et al., 2013*). EpiSCs were cultured on FBS-coated dishes in a chemically defined medium (CDM) supplemented with 20 ng/ml activin A (R&D Systems, Minneapolis, MN) and with 12 ng/ml bFGF (Invitrogen, Carlsbad, CA) (CDM/AF) as described previously (*Brons et al., 2007*; *Zhang et al., 2010a*). To generate ESD-EpiSCs (ESC-derived epiblast stem cells), ESCs or cell aggregates were dissociated into single cells after treatment with 0.05% Trypsin-EDTA at 37°C for 2 min. Individual cells were seeded at a density of $2.0 \times 10^5$ cells per 35-mm dish in CDM/AF. After 6 days, the surviving cells formed large compact colonies. P19 cells were cultured as described previously (*Jin et al., 2009*). Factors and inhibitors, including BMP4 (10 ng/ml, R&D Systems, Minneapolis, MN), Wnt3a (100 ng/ml, R&D Systems, Minneapolis, MN), and CHIR99021 (3 μM, Stemgent, Cambridge, MA), were used.

### Gene knockdown and overexpression

For Pou3f1 knockdown in ESCs, the lentiviral vector pLentiLox 3.7, which expresses shRNA and GFP, was used. A reference shRNA sequence (*Huang et al., 2010b*) was used as a negative control. The control and Pou3f1 shRNA sequences are shown in *Supplementary file 1*. Lentiviral packaging and

cell transfection were performed as described (*Tiscornia et al., 2006*). GFP-positive cells were sorted using a FACS-Aria cell sorter (BD Biosciences, San Jose, CA) and propagated. For stable overexpression, Pou3f1 was cloned into the lentiviral expression vector pFUGW-IRES-EGFP (*Naldini et al., 1996*). The PCR primers used in the cloning are listed in *Supplementary file 1*. The empty vector pFUGW-GFP was used as a negative control. For Pou3f1-inducible overexpression, the Pou3f1-IRES-EGFP fragment was constructed and inserted into the lentiviral vector pLVX-Tight-Puro (Clontech, Mountain View, CA). After co-transfection of pLVX-Tight-Puro-Pou3f1-IRES-EGFP and rtTA lentiviruses for 48 hr, the stable transfection was selected by puromycin (2 µg/ml, Sigma). The culture medium supplemented with Dox (2 µg/ml, Sigma-Aldrich, St. Louis, MO) was used for inducing the overexpression of Pou3f1, and Dox was not added to the control group.

## Immunocytochemistry

Immunocytochemistry was performed as described previously (*Xia et al., 2007*). The mouse monoclonal antibodies included anti-Oct4 (Santa Cruz Biotechnology, Santa Cruz, CA), and anti-Tuj1 (Covance, San Diego, CA). The rabbit polyclonal antibodies included anti-Nestin (Upstate Biotech, Lake Placid, NY), anti-Pax6 (Covance, San Diego, CA), and an anti-Sox1/(2)/3 that preferentially recognize Sox1 and Sox3 over Sox2 (*Okada et al., 2004*; *Tanaka et al., 2004*). Cy3 and Cy5 (Jackson Immunoresearch Laboratories, West Grove, PA) secondary antibodies were used in this study. Fluorescence detection and imaging were performed on a Leica confocal microscope or on an Olympus fluorescence microscope.

## RNA preparation and Q-PCR analysis

Total RNA was extracted from cells using TRIzol reagent (Invitrogen, Carlsbad, CA). Reverse transcription and Q-PCR analysis were performed using an Eppendorf Realplex2 (*Peng et al., 2009*). Primers for Q-PCR analysis are listed in *Supplementary file 1*.

## Whole-mount in situ hybridization

Whole-mount in situ hybridizations were performed as described previously (*Huang et al., 2010a*). The following probes were used: *mPou3f1* (3′ UTR of mouse *Pou3f1* mRNA, PCR-amplified from cDNA), *cPou3f1*, *cSox2*, and *cId1*.

## Mouse chimeric embryo analysis

R1 ESCs constitutively expressing pFUGW-IRES-EGFP were used as the control for visualizing the contribution of the injected cells in vivo. To obtain chimeric embryos, GFP-labeled Pou3f1-KD, Pou3f1-OE, or control ESCs were injected into E2.5 mouse blastocysts respectively, and the cells were then transferred into the uteri of day 2.5 pseudopregnant ICR female mice. For the inducible Pou3f1-overexpresing ESCs, the recipient ICR female mice were fed with Dox (2 mg/ml) in water after blastocyst injection. Mouse embryos were collected at E8.5 to E9.0. After transverse section, the fluorescent signals of embryos were detected by confocal microscope. Our animal experiments are conducted with the highest ethical standards.

## Early chick embryo manipulation

Fertilized eggs (Shanghai Academy of Agricultural Sciences, Shanghai, China) were incubated at 38°C to HH stage 3/3+ (*Hamburger and Hamilton, 1992*). Gene electroporation and new culture were performed as described previously (*New, 1955*; *Voiculescu et al., 2008*). The control vector pCAGGS-IRES-GFP and the *Pou3f1* expression construct pCAGGS-mPou3f1-IRES-GFP were used. Whole-mount immunostaining of GFP was performed as described previously (*Huang et al., 2010a*).

## Luciferase assay

The luciferase assay was described previously (*Jin et al., 2009*). Plasmids were co-transfected in ESCs or in P19 cells in N2B27 medium for 24 hr. f Factor treatment was applied for 10 hr, and then the luciferase activities were measured using a Dual-Luciferase Reporter Assay system (Promega, Madison, WI) with a Turner Design 2020 luminometer.

## Chromatin immunoprecipitation (ChIP)

ChIP assays were performed according to the manufacturer's protocol (Protein A/G Agarose/Salmon Sperm DNA [Upstate Biotech, Lake Placid, NY] and Dynabeads Protein A/G [Invitrogen, Carlsbad, CA]), and detailed procedures were described previously (*Jin et al., 2009*). ChIP was performed with

2 µg antibody against phosphorylated Smad1/5/8 (Cell Signaling) or Pou3f1 (Santa Cruz Biotechnology, Santa Cruz, CA). Normal IgG was used as negative control. Q-PCR was used to amplify various regions of the target gene genome, and primers for ChIP-qPCR are listed in *Supplementary file 1*.

## ChIP-Seq data processing

The high-throughput sequencing was performed by the Computational Biology Omics Core, PICB, Shanghai. The SOAP version 2.20 alignment tool was used to align ChIP-Seq reads to the mouse genome build mm9 (*Li et al., 2009*). Only reads with less than two mismatches that uniquely mapped to the genome were used in subsequent analyses. Using FindPeaks Homer software, Pou3f1 binding peaks with fourfold greater normalized tags were identified in ChIP experiments compared with the control (*Heinz et al., 2010*). We calculated the distance from the peak centers to the annotated transcription start sites (TSS) and then defined the nearest genes as peak-related genes.

## RNA-Seq data processing

Raw reads were mapped to mm9 using the TopHat version 1.4.1 program (*Trapnell et al., 2009*). We assigned FPKM (fragment per kilo base per million) as an expression value for each gene using Cufflinks version 1.3.0 software (*Trapnell et al., 2010*). Then, Cuffdiff software was used to identify differentially expressed genes between treatment and control samples (*Trapnell et al., 2013*). Differentially expressed gene heat maps were clustered by k-means clustering using the Euclidean distance as the distance and visualized using Java TreeView software (*Saldanha, 2004*).

## Functional enrichment analysis

To investigate the functions of genes with *Pou3f1* binding sites and differentially expressed after *Pou3f1* perturbation, functional enrichment analyses were performed using the Database for Annotation, Visualization, and Integrated Discovery (DAVID).

## Statistics

Each experiment was performed at least three times, and similar results were obtained. The data are presented as the mean ± SD. Student's *t* test was used to compare the effects of all treatments. Statistically significant differences are indicated as follows: * for $p < 0.05$ and ** for $p < 0.01$.

## Acknowledgements

We thank Dr Michael Wegner (Universitat Hamburg, Germany) for the *mPou3f1* plasmid, Dr Dies Meijer (Erasmus University Rotterdam, Netherland) for the *cPou3f1* plasmid, Dr Hisato Kondoh (Osaka University, Japan) for the *Sox2* enhancer plasmids, and Dr Claudio Stern (University College London, UK) for the *cSox2* chicken probes and for the *Xenopus* BMP4 expression plasmid.

## Additional information

### Funding

| Funder | Grant reference number | Author |
|---|---|---|
| Chinese Academy of Sciences | Strategic Priority Research Program, XDA01010201 | Naihe Jing |
| National Natural Science Foundation of China | 91219303 | Naihe Jing |
| Ministry of Science and Technology of the People's Republic of China | National Key Basic Research and Development Program of China, 2014CB964804 | Naihe Jing |

The funder had no role in study design, data collection and interpretation, or the decision to submit the work for publication.

### Author contributions

QZ, LS, GP, Conception and design, Acquisition of data, Analysis and interpretation of data, Drafting or revising the article; NS, KT, Analysis and interpretation of data, Drafting or revising the article; JC, TZ, WT, CQ, Acquisition of data; NS, Conception and design, Acquisition of data;

YQ, J-DJH, JL, Analysis and interpretation of data; NJ, Conception and design, Analysis and interpretation of data, Drafting or revising the article

### Ethics

Animal experimentation: This study was performed in strict accordance under the ethical guidelines of the Institute of Biochemistry and Cell Biology and all experiments were approved by the committee on the Ethics of Animal Experiments of the Shanghai Institute of Biochemistry and Cell Biology.

## Additional files

### Supplementary file

• Supplementary file 1. Primer list for PCR analysis. (**A**) The PCR primers used to clone Pou3f1 into the lentiviral expression vector pFUGW-IRES-EGFP. (**B**) Oligo sequences used for Pou3f1 RNAi. (**C**) Primers used for Real-time Q-PCR analysis. (**D**) Primers used for ChIP-qPCR.

### Major dataset

The following dataset was generated:

| Author(s) | Year | Dataset title | Dataset ID and/or URL | Database, license, and accessibility information |
|---|---|---|---|---|
| Qingqing Zhu, Lu Song, Guangdun Peng, Na Sun, Jun Chen, Ting Zhang, Nengyin Sheng, Wei Tang, Cheng Qian, Yunbo Qiao, Ke Tang, Jing-Dong Jackie Han, Jinsong Li, Naihe Jing | 2014 | Data from: Pou3f1 promotes neural fate commitment via activation of neural lineage genes and inhibition of BMP/Wnt signals | doi: 10.5061/dryad.3vk1g | Available at Dryad Digital Repository under a CC0 Public Domain Dedication. |

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
