## [Decision Letter]

Thank you for sending your work entitled “Oct6 promotes neural commitment via activation of neural lineage genes and inhibition of BMP/Wnt signals” for consideration at *eLife*. Your article has been favorably evaluated by a Senior editor, a Reviewing editor, and 3 reviewers, one of whom, Ali Brivanlou, has agreed to reveal his identity.

The Reviewing editor and the reviewers discussed their comments before we reached this decision, and the Reviewing editor has assembled the following comments to help you prepare a revised submission.

In this manuscript, the authors analyzed the role of Oct6 in during neuronal commitment and differentiation by using mouse ES cell differentiation system. The three reviewers' comments are mostly positive for this manuscript. However, to be acceptable, we would urge authors to make substantial revisions as summarized below.

1) Authors need to clarify what they mean by “neural commitment”. In many cases, very vague words are used in the text to describe the role of Oct6, e.g. “essential positive factor” or “critical regulator”, etc.

2) At the beginning of the paper, while authors are referring some microarray time course data, such data are not represented in Figure 1. Furthermore, no reference was added in the text. Authors need to specify this.

3) While authors claim that Oct6 expression is detectable in undifferentiated ES cells and increases during differentiation. Then, what is the role of Oct6 in ES cells? Does Oct6 bind to the Wnt/BMP target genes and repress them?

4) Controls for overexpression and KD experiments are not clear. Authors need to clarify it. Furthermore, the effects of Oct6 KD and overexpression are relatively small upon gene expression profile and cell fate than expected. How do such relatively small changes translate to the role of Oct6 in neural induction by targeting both intrinsic transcription factors and extracellular signaling molecules? Authors need to explain about these results.

5) In the Oct6 KD experiment, there is no information about the level of KD. It is necessary to show how much the shRNAs to Oct6 is down-regulating the Oct6 gene at the transcript and protein level.

6) Do Oct-family members compensate for the depletion of Oct6?

7) It also needs to be shown that constitutive and dox-inducible Oct6 overexpression in the mouse ES cells in within the physiological range at the transcript and protein levels.

8) The Oct6 target genes identified in gain-of-function lines should be validated in the loss-of-function lines.

9) Multiple media conditions are used in various parts of the manuscript during neural differentiation. For example, data in Figure 1 is not compatible to data from other figures. Ideally, the data for all figures should be presented in serum free conditions.

10) As regards the results of Oct6 loss-of-function experiments are not consistent with those obtained in the previous report (Iwabuchi-Doi et al., 2012). Such differences should be explained in the text.

---

## [Author Response]

*1) Authors need to clarify what they mean by “neural commitment”. In many cases, very vague words are used in the text to describe the role of Oct6, e.g. “essential positive factor” or “critical regulator”, etc*.

To clarify the description, we replaced “neural commitment” with “neural fate commitment” in the revised manuscript (e.g., page 2, line 29). Neural fate commitment is an essential step in neural induction, which is defined as the process by which cells differentiate to the neural fate even in the presence of inhibitory signals (57). In addition, we also removed the words mentioned by the reviewers, such as “essential positive factor,” and “critical regulator” from the revised manuscript.

*2) At the beginning of the paper, while authors are referring some microarray time course data, such data are not represented in*
Figure 1*. Furthermore, no reference was added in the text. Authors need to specify this*.

To identify the intrinsic factors that are involved in the neural differentiation of pluripotent stem cells, we performed microarray assays in mouse ESCs undergoing neural differentiation. ESCs were aggregated as embryonic bodies in 8% knockout serum replacement medium from days 0 to 6 (Zhang et al., 2010). RNA was extracted from ESCs at days 0, 2, 4, and 6 samples, and analyzed using Agilent Whole Mouse Genome Oligo 4X44K microarrays. The microarray data have not been published yet, and we have uploaded the data to the Dryad (doi:10.5061/dryad.3vk1g). Using differential gene expression (DEG) analysis, we identified multiple genes that were up-regulated during ESC neural differentiation, including *Pou3f1***.**

3) While authors claim that Oct6 expression is detectable in undifferentiated ES cells and increases during differentiation. Then, what is the role of Oct6 in ES cells? Does Oct6 bind to the Wnt/BMP target genes and repress them?

We thank the reviewers for raising this interesting question. Pou3f1 expression is detectable in undifferentiated ES cells, indicating that *Pou3f1* might be involved in ESC pluripotency maintenance. However, Pou3f1 expression in undifferentiated ESCs is much lower compared with differentiated ESCs at days 2 and 4 (Figure 1 and Figure 1—figure supplement 1). In addition, the expression profiles of pluripotency markers were similar in Pou3f1 knockdown ESCs and in control ESCs (Figure 1—figure supplement 1). Therefore, the role of Pou3f1 in ESCs is not obvious.

We also examined whether altering Pou3f1 expression in ESCs affects the expression of Wnt/BMP target genes. As shown in Figure 8 below, both knockdown and overexpression of *Pou3f1* have no effect on the expression of Wnt/BMP target genes in ESCs, indicating that Pou3f1 most likely does not bind to and repress the Wnt/BMP target genes in ESCs.Author response image 1.

*4) Controls for overexpression and KD experiments are not clear. Authors need to clarify it. Furthermore, the effects of Oct6 KD and overexpression are relatively small upon gene expression profile and cell fate than expected. How do such relatively small changes translate to the role of Oct6 in neural induction by targeting both intrinsic transcription factors and extracellular signaling molecules? Authors need to explain about these results*.

We apologize for not making this clear in the original manuscript. For the Pou3f1-inducible overexpression experiment, we used no doxycycline addition group as control. For stable overexpression cell lines, mPou3f1 was inserted into the lentiviral expression vector pFUGW-IRES-EGFP (33). The empty lentiviral expression vector pFUGW-GFP was used as a negative control. For the knockdown experiments, the lentiviral vector pLentiLox 3.7, which expresses a control shRNA sequence (Huang et al., 2010), was used as a negative control. We have added this information to the Materials and methods section in the revised manuscript.

For the *Pou3f1* knockdown experiment, we found that the POU III family member *Brn2* could compensate for Pou3f1 depletion, and we will discuss this finding in further detail in Question 6. Briefly, we found that *Brn2* expression increases after *Pou3f1* knockdown, whereas the simultaneous knockdown of *Pou3f1* and *Brn2* decreases the expression of neural markers more significantly (Figure 1—figure supplement 2). Thus, the relatively small effects of Pou3f1 knockdown are likely due to Brn2 compensation.

As an intrinsic neural fate promoting factor, *Pou3f1* is expressed earlier than the neural lineage related genes *Zfp521*, *Pax6*, and *Zic1*, and *Pou3f1* directly binds and upregulates these genes expression (Figures 1 and 4). Pou3f1-dependent direct regulation of the So*x2*N2 enhancer serves as additional support for its neutralization function. These factors synergistically contribute to neuroectoderm formation (17); Pou3f1 controls their expression intrinsically and then promotes early neural fate commitment. On the other hand, the repression of extrinsic signals is essential for neural fate commitment. We demonstrate that the Pou3f1-dependent repression of BMP and Wnt signaling genes occurs at the transcriptional level (Figure 6). Pou3f1 coordinates the expression of both endogenous factors and exogenous signals at the transcriptional level, which is a fast and efficient approach to regulating cell fate determination.

*5) In the Oct6 KD experiment, there is no information about the level of KD. It is necessary to show how much the shRNAs to Oct6 is down-regulating the Oct6 gene at the transcript and protein level*.

We examined the effect of shRNAs to Pou3f1 on the transcript and protein levels, and found that shRNA1 (Pou3f1-KD1) and shRNA3 (Pou3f1-KD3) decreases the expression of *Pou3f1* transcripts in ESCs by approximately 50% and 30%, respectively; however, shRNA2 (Pou3f1-KD2) does not affect *Pou3f1* expression in ESCs (Figure 1—figure supplement 1). Similar results were observed regarding the expression of Pou3f1 protein with various Pou3f1-shRNAs via Western blot (Figure 1—figure supplement 1). We have added this information to the revised manuscript.

6) Do Oct-family members compensate for the depletion of Oct6?

This question is important. Oct-family members belong to POU domain factors. In previous studies, the functional redundancy between various POU factors has been reported ([1]: [8]; [18]). Thus, we sought to address whether the compensation by other POU III family members, Brn1 and Brn2, plays a role in Pou3f1 depletion. First, we examined the expression patterns of these genes during ESC neural differentiation and observed that Brn1 and Brn2 are up-regulated in serum-free medium after day 5 (Figure 1—figure supplement 2), which occurs much later than Pou3f1. Consistently, in mouse embryos *in vivo*, Pou3f1 expression is detected at E5.5, which occurs much earlier than neural induction in the mouse embryo (Figure 3) (66). In addition, Brn1 and Brn2 expression is only detected after E8.5 (Maxime Bouchard et al., 2005). Brn1 and Brn2 expression patterns in ESCs *in vitro* and mouse embryos *in vivo* suggest that distinct critical windows potentially exist for each POU factor to modulate neural developmental events.

Next, we assessed whether Pou3f1 depletion affects Brn1 and Brn2 expression. Indeed, Brn2 expression is obviously up-regulated in Pou3f1-KD1 and Pou3f1-KD3 cell lines compared with the control cells, whereas Brn1 expression is unchanged (Figure 1—figure supplement 2). In addition, when both Pou3f1 and Brn2 were simultaneously knocked-down by co-transfection with lentiviral-mediated shRNAs, Brn1 expression was not affected (Figure 1—figure supplement 2). Then, the expression of neural markers decreased more dramatically in the Pou3f1 and Brn2 double-knockdown cells compared with control or Pou3f1 knockdown cells (Figure 1—figure supplement 2). This information was added to the revised manuscript. Together, these results suggest that Brn2 potentially compensates for Pou3f1 depletion.

*7) It also needs to be shown that constitutive and dox-inducible Oct6 overexpression in the mouse ES cells in within the physiological range at the transcript and protein levels*.

The gene functional analysis is achieved by loss-of-function or gain-of-function. Typically, a gain-of-function analysis in embryos and tissues is executed through the ectopic expression of a gene within the physiological range. However, in the cultured cells, gain-of-function is performed by overexpression, which will lead the gene expression level higher than the physiological range. To answer this question, we examined *Pou3f1* expression patterns during ESC neural differentiation in control, Dox-inducible and in Pou3f1-stable overexpression cell lines. We found that the *Pou3f1* expression peaked at days 2-4 of ESC neural differentiation in control cells (Figure 9), which is consistent with the data in Figure 1 and Figure 1—figure supplement 1. During the same period, Pou3f1 expression levels in the Dox-inducible and stable lines were approximately two-fold higher compared with control cells; except that Dox-induced Pou3f1 expression displayed a considerably higher peak at day 1 (Figure 9). After day 5, all three cell lines displayed similar *Pou3f1* expression levels.Author response image 2.

*8) The Oct6 target genes identified in gain-of-function lines should be validated in the loss-of-function lines*.

As suggested by the reviewers, the Pou3f1 target genes identified in gain-of-function lines have been validated in the loss-of-function lines. The expression of neural lineage-related genes, such as *Sox2*, *Zfp521*, *Zic1*, and *Zic2*, increased in *Pou3f1* overexpressing ESCs (Figure 5), and their expression decreased after *Pou3f1* depletion (Figure 5). The expression of BMP and Wnt pathway-target genes, such as *Id1/2*, *Msx1/2*, *Wnt3*, *Axin2*, *Dkk1*, and *Myc*, was reduced in *Pou3f1* overexpressing ESCs (Figure 6), and their expression increased after *Pou3f1* knockdown (Figure 6).

*9) Multiple media conditions are used in various parts of the manuscript during neural differentiation. For example, data in*
Figure 1
*is not compatible to data from other figures. Ideally, the data for all figures should be presented in serum free conditions*.

In this study, we used two culture conditions for ESC neural differentiation: serum-free medium (8% knockout replacement serum) and serum-containing medium (10% FBS). In serum-free medium, the percentage of Sox^+^/Oct4^-^ neural progenitor cells (NPCs) can reach 80% of total cells by day 6 (Zhang et al., 2010); these conditions are not optimal for a gain-of-function analysis of neural fate-promoting factors but are suitable for a loss-of-function analysis. In contrast, ESC neural differentiation was inhibited in serum-containing medium, which contains BMP and other neural inhibitory signals. In addition, the percentage of NPCs is considerably reduced compared with serum-free medium after 8 days of differentiation (data not shown). The serum-containing medium is more suitable for the gain-of-function analysis of neural fate-promoting factors. Thus, we performed Pou3f1 knockdown experiments in serum-free medium (Figure 1) and Pou3f1 overexpression assays in serum-containing medium (Figure 1). The culture condition used depends on the experimental approaches and effects. We have added this culture condition information to the revised manuscript.

As suggested by the reviewers, we also performed the Pou3f1 overexpression experiments in serum-free medium to further confirm the function of Pou3f1 in ESC neural differentiation. The results indicate that the NPC markers, such as Sox1, Pax6, and Nestin, and the neuron markers Tuj1 and Map2 were notably upregulated in the day 4 EBs (Figure 1—figure supplement 3). The immunostaining assays also revealed an increased number of Sox^+^/Oct4^-^ NPCs, Nestin^+^ NPCs, and Tuj1^+^ neurons in the Pou3f1-stable overexpressing cells at day 4 EBs compared with control cells (Figure 1—figure supplement 3). Cells in EBs from various days were replated in N2 medium for neuronal differentiation, and a significantly increased number of Tuj1^+^ neurons were observed at day 4 in the Pou3f1-overexpressing ESCs (Figure 1—figure supplement 3d). These results suggest that ESC neural differentiation is accelerated by Pou3f1 overexpression in serum-free culture. We have added this information to the “Results” section of Figure 1 in the revised manuscript.

*10) As regards the results of Oct6 loss-of-function experiments are not consistent with those obtained in the previous report (Iwabuchi-Doi et al., 2012). Such differences should be explained in the text*.

In the previous report, Iwabuchi-Doi et al. found that *Pou3f1* was expressed at E7.5-E7.75 in the anterior portion of mouse embryo (17), which contributes to neuroectoderm formation (43). They showed that *Pou3f1* overexpression promotes the expression of neural lineage marker genes, such as *Sox1* and *Pax6*, and concluded that *Pou3f1* was involved in anterior neural plate development. In fact, our observations in Pou3f1 gain-of-function assays (Figure 1 and Figure 1—figure supplement 3) are consistent with their findings. The discrepancy between Iwabuchi-Doi’s study and ours occurs in the Pou3f1 loss-of-function assays. Iwabuchi-Doi et al. did not report Pou3f1 knockdown results; however, we demonstrated that Pou3f1 is required for ESC neural differentiation (Figure 1). Several possibilities may contribute to the discrepancy between our study and Iwabuchi-Doi’s:

First, we differentiated mouse ESCs and EpiSCs in cell aggregates (EBs) in serum-free medium, whereas Iwabuchi-Doi et al. performed EpiSC neural differentiation using monolayers in N2B27 medium. Thus, the different differentiation protocols may account for the different observations between these two studies.

Second, the POU III family member *Brn2* compensates for *Pou3f1* depletion in ESC neural differentiation (Figure 1—figure supplement 2); therefore, it is difficult to capture the phenotype of ESC neural differentiation after *Pou3f1* knockdown.

Third, Iwabuchi-Doi et al. have studied multiple transcriptional factors to establish the transcriptional regulatory networks during the anterior neural plate development, and *Pou3f1* is only one of six factors that these authors analyzed. In our study, we exclusively focused on the function and mechanism of *Pou3f1* during ESC neural differentiation, and we have to confirm Pou3f1’s function in the loss-of-function experiment very carefully. Given that the exon sequence of the *Pou3f1* gene is GC-rich, it is extremely difficult to knockdown its expression effectively. We tried several shRNAs against *Pou3f1*, but only two of the shRNAs (Pou3f1-shRNA1 and Pou3f1-shRNA3) worked. We used these two shRNAs in our study.

We have explained this discrepancy in the revised manuscript.